# Diet-responsive transcriptional regulation of insulin in a single neuron controls systemic metabolism

**Ava Handley[1]\*, Qiuli Wu[1,2], Tessa Sherry[1], Rebecca Cornell[1], Roger Pocock[1]\***

**1** Development and Stem Cells Program, Monash Biomedicine Discovery Institute and Department of Anatomy and Developmental Biology, Monash University, Melbourne, Australia, **2** Key Laboratory of Developmental Genes and Human Diseases in Ministry of Education, Medical School of Southeast University, Nanjing, China

\* ava.handley@monash.edu (AH); roger.pocock@monash.edu (RP)

**Data Availability Statement:** All relevant data are within the paper and its Supporting Information files.

## Abstract

Metabolic homeostasis is coordinated through a robust network of signaling pathways acting across all tissues. A key part of this network is insulin-like signaling, which is fundamental for surviving glucose stress. Here, we show that *Caenorhabditis elegans* fed excess dietary glucose reduce insulin-1 (INS-1) expression specifically in the BAG glutamatergic sensory neurons. We demonstrate that INS-1 expression in the BAG neurons is directly controlled by the transcription factor ETS-5, which is also down-regulated by glucose. We further find that INS-1 acts exclusively from the BAG neurons, and not other INS-1-expressing neurons, to systemically inhibit fat storage via the insulin-like receptor DAF-2. Together, these findings reveal an intertissue regulatory pathway where regulation of insulin expression in a specific neuron controls systemic metabolism in response to excess dietary glucose.

## Introduction

All organisms must adapt their energy use in order to maintain homeostasis in the face of an ever-changing environment. The nervous system is the master regulator of energy metabolism—integrating metabolic information from peripheral tissues, and driving shifts in resource allocation, modulating thermogenesis, adapting nutrient absorption, and altering feeding behavior. Insulin signaling is a key metabolic regulator that stimulates the uptake of excess blood glucose to be stored as glycogen and fat [1]. Beyond regulating blood glucose, insulin signaling in the nervous system controls diverse processes such as behavior, reproduction, neurotransmission, neuroplasticity, development, and aging [2–10]. The *Caenorhabditis elegans* genome contains 40 insulin-like peptide genes, most of which are neuronally expressed [11,12]. Despite the large number of insulin-like peptides, there is only a single identified ortholog of the insulin receptor, DAF-2, which is broadly expressed across all tissues [6]. DAF-2 is essential for enabling the worm to integrate metabolism with developmental decisions [6,8,13–16]. When DAF-2 function is mildly reduced, animals store more fat and reduce their

**Funding:** This work was supported by the National Health and Medical Research Council (GNT1105374 and GNT1137645) and a veski Innovation Fellowship (VIF23) to R.P. The funders had no role in study design, data collection and analysis, decision to publish, or preparation of the manuscript.

**Competing interests:** The authors have declared that no competing interests exist.

**Abbreviations:** CTCF, calculated total fluorescence; GFP, green fluorescent protein; MUFA, mono-unsaturated fatty acid; ORO, Oil-Red O; PUFA, poly-unsaturated fatty acid; RFP, red fluorescent protein; RNAi, RNA interference; SFA, saturated fatty acid; TAG, triglyceride.

food-searching behavior [6,8,17–19]. In contrast, when DAF-2 function is severely reduced, animals again store more fat, but alter their developmental trajectory toward a long-lived, stress-resistant dauer stage [20]. Much like humans, *C. elegans* that consume a diet supplemented with high glucose have increased fat deposition [21,22] and reduced life span [23–26], and insulin-like signaling is embedded in this physiological response [23,26,27].

Based on sequence, structural predictions and the presence of a cleaved C-peptide, insulin-1 (INS-1) is the closest *C. elegans* ortholog to human insulin [12]. INS-1 action is diverse, with roles in learning, memory, and development [28–33]. How INS-1 affects the DAF-2 receptor is context specific, for example, INS-1 inhibits DAF-2 to promote the starvation-induced dauer state [12,32], and activates DAF-2 during salt-chemotaxis learning [30]. Despite being the closest insulin ortholog, it had not been established whether INS-1 plays a role in the adaptive response to a high glucose diet, and promotes fat storage like its mammalian counterpart.

In this study, we aimed to determine whether INS-1, acting from the nervous system, forms part of the metabolic response to excess glucose in *C. elegans*. Consuming a glucose-rich diet can have a devastating effect on an organism's health. *C. elegans* studies have used a range of glucose concentrations, with a very high glucose diet (approximately 111 mM or 2%) over the lifetime of the animal being frequently used [21,26]. However, excessive glucose consumption of over 80 mM during development can cause severe damage to mitochondria and elicit oxidative stress [25,34,35]. To circumvent these developmental and physiological defects caused by a long-term high glucose diet, we assessed INS-1 action in response to short-term, relatively low glucose exposure (40 mM or 0.7%), at a late developmental stage. Under these conditions, we find that *ins-1* expression is down-regulated in a pair of glutamatergic sensory neurons, the BAG neurons. Down-regulation of *ins-1* in response to excess glucose does not require neurotransmission, neuropeptide signals, or cilia function, and also occurs in response to nonmetabolizable L-glucose. Further, we find that *ins-1* expression in the BAG neurons is directly regulated by ETS-5, a transcription factor necessary for BAG-fate specification [36,37]. We show that under standard food conditions, *ins-1* acts specifically from the BAG neurons to inhibit fat levels in intestinal cells, which, in turn, promotes food-seeking behavior. Finally, we show that INS-1 activates the insulin-like receptor DAF-2 in the intestine to regulate fat storage through control of the FOXO transcription factor DAF-16. Together, our study reveals a mechanism whereby the ETS-5 transcription factor modulates *ins-1* expression in the postmitotic BAG neurons in response to changes in dietary glucose. This neuronal signaling mechanism enables metabolic and behavioral adaptation in response to nutritional challenges in distal tissues.

## Results

### Excess dietary glucose down-regulates *ins-1* specifically in the BAG neurons

To analyze *ins-1*-expression dynamics, we generated a transcriptional reporter transgenic line using a 2.5 kb fragment of the *ins-1* promoter driving nuclear-localized green fluorescent protein (GFP) (Fig 1A). This *ins-1p::NLS-GFP* transcriptional reporter enables clear visualization of INS-1 positive neurons, which we identified by neuron position, previously published expression patterns [12] and single-cell transcriptomic data [38,39] (Figs 1A, 1B, S1A and S1B). To determine whether *ins-1* promoter activity is affected by excess dietary glucose, we fed mid-L4 stage *ins-1p::NLS-GFP* larvae *Escherichia coli* OP50 + 40 mM glucose for 24 hours, and assayed NLS-GFP levels in each identifiable neuron. We found that *ins-1* promoter activity in most *ins-1* positive neurons did not respond to this glucose-enriched diet (Figs 1B, 1C, S1C and S1D). However, we observed a significant decrease (approximately 50%) in *ins-1p*::

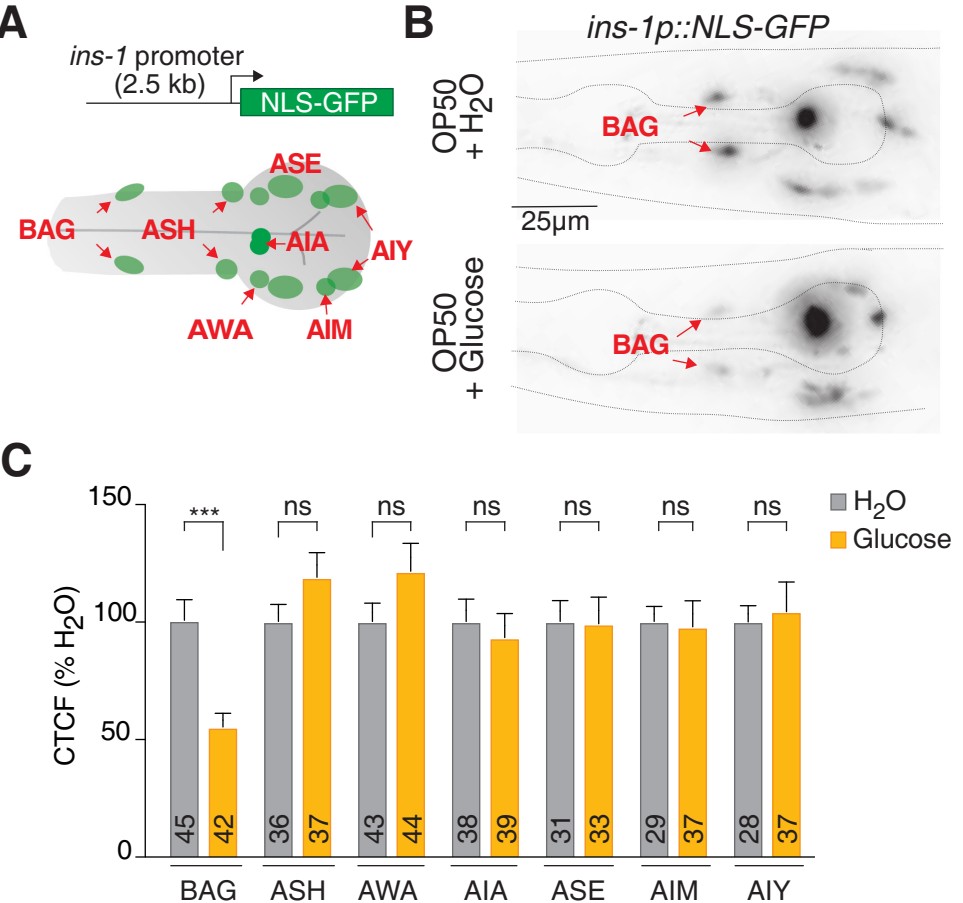

**Fig 1. A high glucose diet down-regulates *ins-1* promoter activity in the BAG neurons.** (**A**) A 2.5-kb fragment of the ***ins-1* promoter** drives expression in >10 neurons. (**B**) Representative images of *ins-1p*::*NLS-GFP* expression in *E. coli* OP50-fed (upper panel) and *E. coli* OP50 + 40 mM glucose-fed (lower panel) animals (BAG neurons indicated by red arrows). Diet treatment: 24 hours. Scale bar 25 µm. (**C**) Quantification of nuclear GFP intensity across all observable neurons of the *ins-1p*::*NLS-GFP* reporter grown on *E. coli* OP50 or *E. coli* OP50 + 40 mM glucose. Diet treatment: 24 hours. Data presented as CTCF % of no glucose, $\bar{x}$ + SEM, *n* shown within bars, ns, not significant ($p > 0.05$), *** = $p \leq 0.001$, significance assessed by unpaired *t* test. The underlying numerical data can be found in S1 Data. CTCF, calculated total fluorescence; GFP, green fluorescent protein.

*NLS-GFP* expression in the BAG glutamatergic sensory neurons in response to the excess dietary glucose (Fig 1B and 1C).

There are several ways that excess dietary glucose may down-regulate *ins-1* expression in the BAG neurons. The effect of glucose on OP50 growth, the BAGs or other neurons may sense glucose in the external environment, other neurons that sense metabolic changes may signal to the BAG neurons to regulate *ins-1* expression, the BAG neurons may autoregulate *ins-1* expression via autocrine signaling or the BAG neurons may directly sense internal increases in glucose. To test the effect of glucose on OP50 growth, we measured BAG *ins-1p*::*NLS-GFP* expression when OP50 was heat-killed prior to adding to the glucose plates. We found that BAG *ins-1p*::*NLS-GFP* expression was significantly decreased on the heat-killed bacterial plates (Fig 2A); therefore, alterations to OP50 growth or metabolites when grown on glucose do not contribute to BAG-*ins-1* down-regulation. We then determined whether external sensing was involved in down-regulating BAG *ins-1p*::*NLS-GFP* expression using a *che-3* mutant that abolishes cilia structure and function in all ciliated neurons [40,41]. In this

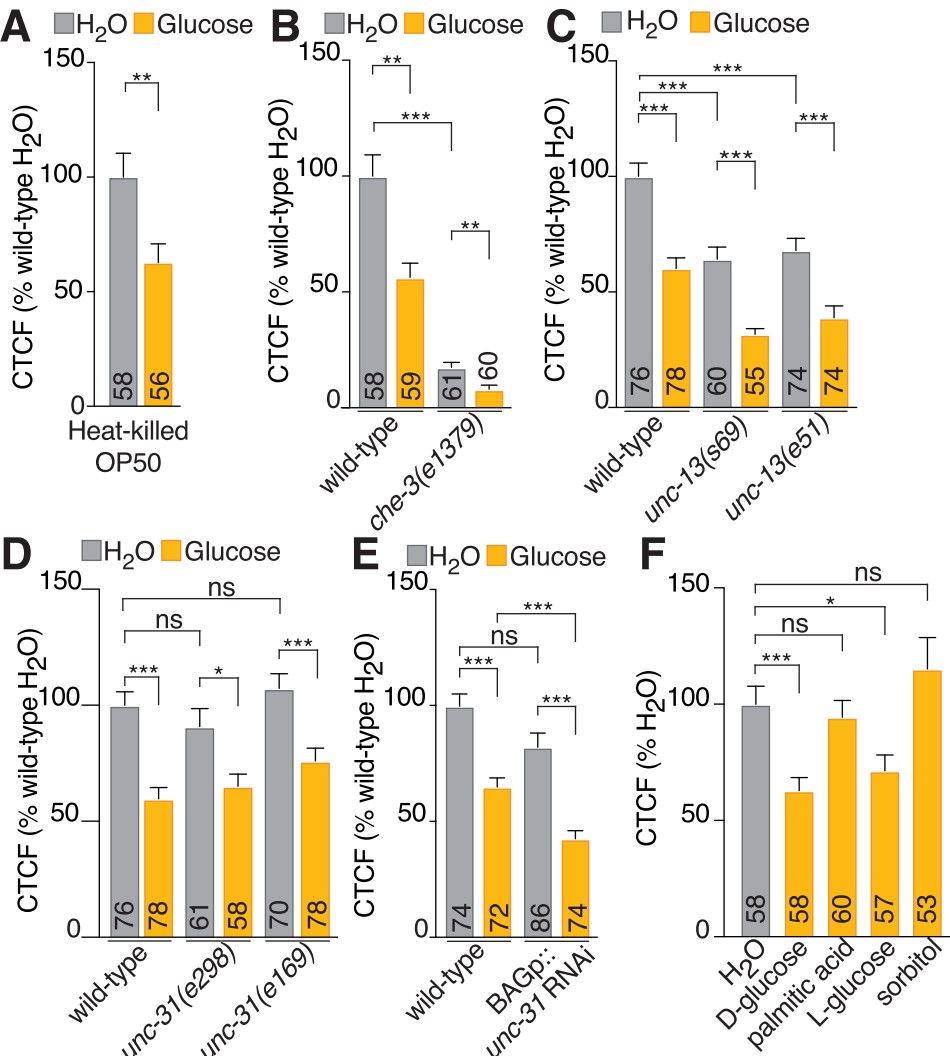

**Fig 2. Excess glucose acts directly on the BAG neurons to down-regulate *ins-1*.** Quantification of nuclear GFP intensity in BAG neurons of the *ins-1p::NLS-GFP* reporter in **(A)** OP50 that was heat-killed prior to plating on $H_2O$ or glucose plates, **(B)** mutant for cilia formation (*che-3*), **(C)** mutants for neurotransmission (*unc-13*), **(D)** mutants for neuropeptide secretion (*unc-31*), and **(E)** animals with BAG-specific RNAi-mediated knock-down of *unc-31*, grown on *E. coli* OP50 or *E. coli* OP50 + 40 mM glucose. **(F)** Quantification of nuclear GFP intensity in BAG neurons of the *ins-1p::NLS-GFP* reporter grown on D-glucose, palmitic acid, L-glucose, and D-sorbitol. Diet treatment: 24 hours. Data presented as CTCF % of wild-type on standard diet ($H_2O$), $\bar{x}$ + SEM, $n$ shown within bars, ns, not significant ($p > 0.05$), * = $p \leq 0.05$, ** = $p \leq 0.01$, *** = $p \leq 0.001$, A–D significance assessed by mixed-effects model (2-way ANOVA). E significance assessed by 1-way ANOVA with Dunnet's correction. The underlying numerical data can be found in S1 Data. CTCF, calculated total fluorescence; GFP, green fluorescent protein; RNA interference.

mutant, the BAGs and other ciliated neurons have no connection to, and therefore cannot sense, the external environment. While *ins-1p::NLS-GFP* expression was diminished in the *che-3(e1379)* mutant compared to wild-type animals on the standard diet, *ins-1* promoter activity in the BAGs was still significantly reduced in *che-3* mutant animals on excess glucose (Figs 2B and S2A). We next tested whether signaling from other neurons to the BAG neurons controls *ins-1* expression in response to excess dietary glucose. Neurons use 2 broad means of communication to other neurons and tissues—neurotransmission and neuropeptide signaling [42]. We found that *unc-13* mutants, which are defective for neurotransmission [43], showed

significantly lower *ins-1* expression compared to wild-type animals on the standard diet, but *ins-1p::NLS-GFP* expression was significantly down-regulated in response to excess dietary glucose (Figs 2C and S2A). Abolishing neuropeptide secretion using *unc-31* mutants [44] did not significantly affect *ins-1* expression levels on a standard diet, and these mutants were still able to significantly down-regulate *ins-1* expression on excess dietary glucose (Figs 2D and S2A). Autocrine insulin-like signaling is important for establishing correct neuropeptide expression during BAG neuron development [45]. We generated an integrated transgenic line that knocks down *unc-31* specifically in the BAG neurons using the BAG-specific *gcy-33* promoter to drive double-stranded *unc-31* RNA (*BAGp::unc-31 RNAi*). We found that BAG-specific *unc-31* knockdown did not affect the response of the *ins-1p::NLS-GFP* reporter to excess dietary glucose (Figs 2E and S2A). Together, these data suggest that the BAG neurons receive a direct, internal signal after consuming excess glucose that leads to *ins-1* down-regulation. To determine what type of signal is received by the BAG neurons, we tested the response of the *ins-1p::NLS-GFP* reporter to different compounds. First, we fed worms palmitic acid (C16:0), a saturated fatty acid that can be used by worms to generate other types of fatty acid [46]. We found animals fed palmitic acid had the same BAG neuron *ins-1* expression level as animals on a standard diet (Figs 2F and S2B). Next, we fed *ins-1p::NLS-GFP* animals with L-glucose, the nonmetabolizable enantiomer of D-glucose. We found that L-glucose, like D-glucose, reduced BAG *ins-1* expression (Figs 2F and S2B). One possibility is that the BAG neurons are responding to a change in osmolarity that occurs with excess glucose in the diet. To test whether this was the case, we fed *ins-1p::NLS-GFP* animals with the nondigestible sugar sorbitol, which is expected to induce a change in osmolarity comparable to glucose without the nutritional value. We found no change to BAG *ins-1* expression in sorbitol-fed animals compared to standard diet (Figs 2F and S2B). These data suggest that glucose itself, and not a downstream metabolite derived from glucose or a change in osmolarity, causes down-regulation of *ins-1* in the BAG neurons.

## ETS-5 directly regulates *ins-1* expression in the BAG neurons

To identify how *ins-1* is transcriptionally regulated by glucose specifically within the BAG neurons, we examined the *ins-1* promoter for putative transcription factor motifs [47]. We identified two putative ETS-binding sites that occur 130 bp apart and approximately 1 kb upstream of the *ins-1* start codon (Fig 3A). Interestingly, the E-26(ETS)-domain transcription factor FEV/Pet1 regulates insulin expression in mouse pancreatic β-cells [48]. Furthermore, the *C. elegans* ortholog of FEV/Pet1, ETS-5, is a key regulator of BAG-fate specification and controls intestinal fat storage, metabolism, and feeding behavior [49,50]. Therefore, ETS-5 was a clear candidate regulator of *ins-1* transcription in the BAG neurons in response to excess glucose. To determine if *ins-1* expression in the BAG neurons is regulated by ETS-5, we combined the *ins-1* transcriptional reporter line with the *ets-5(tm1734)* mutant. We found that *ins-1* promoter activity in *ets-5(tm1734)* animals was significantly diminished in the BAG neurons (Fig 3B and 3C). We did not detect any overt changes to *ins-1* expression levels in other *ins-1* positive neurons in the *ets-5* mutant. To demonstrate this, we quantified *ins-1* expression in the ASH neurons, which have similar *ins-1* expression levels and are on a similar focal plane to the BAG neurons. We found that *ins-1* expression in the ASH neurons did not change in the *ets-5* mutant compared to wild-type worms (Fig 3B and 3C). That *ins-1* expression is only regulated in the BAG neurons by ETS-5 is consistent with the observation that *ets-5* and *ins-1* expression only colocalises in the BAG neurons and no other neurons (S3A Fig).

ETS-5 is a candidate terminal selector transcription factor in the BAG neurons, meaning that the observed *ins-1* down-regulation may be an indirect effect of removing ETS-5 activity.

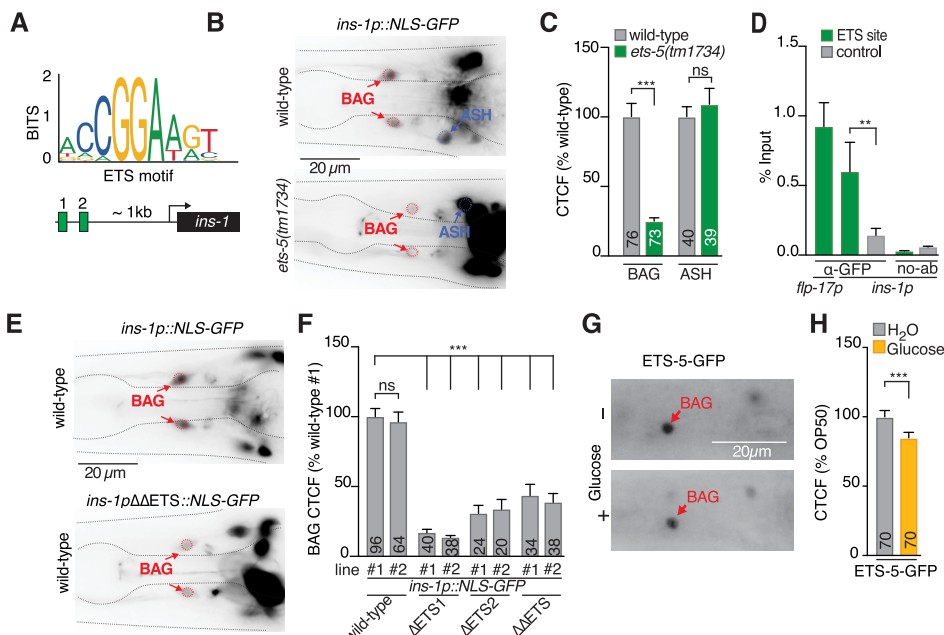

**Fig 3. The BAG-specifying transcription factor ETS-5 regulates *ins-1* expression.** (**A**) The *ins-1* promoter contains two adjacent ETS motifs (green boxes). (**B**) Representative images of *ins-1p*::*NLS-GFP* expression in *ets-5(tm1734)* animals compared to wild-type. Red circles: BAG neurons, blue circle: ASH neurons. Scale bar 20 μm. (**C**) Quantification of GFP intensity (CTCF) of *ins-1p*::*NLS-GFP* in the BAG and ASH neurons of wild-type and *ets-5 (tm1734)* animals. Data presented as $\bar{x}$ + SEM, *n* shown within bar. Significance assessed by unpaired *t* test. (**D**) ETS-5-GFP ChIP-qPCR. Regions assessed: *flp-17* promoter and *ins-1* promoter containing ETS-sites (green), *ins-1* promoter non-ETS-5 site region (gray), and no-antibody control (no-ab). Data are presented as $\bar{x}$ +SEM, *n* = 4. Significance assessed by ratio paired *t* test. (**E**) Representative images of *ins-1p*::*NLS-GFP* with both putative ETS sites mutated (ΔΔETS) compared to wild-type. Red circles: BAG neurons. Scale bar 20 μm. (**F**) Quantification of GFP intensity (CTCF) in BAG relative to wild-type control line #1, with *ins-1* promoter mutated at ETS1, ETS2, or both, 2 independent lines assayed, *n* shown in bar. Significance assessed by 1-way ANOVA with Dunnet's correction. (**G**) Representative images of endogenous ETS-5-GFP expression in the BAG neurons (red arrow) of animals fed standard diet (*E. coli* OP50) (upper panel) and high glucose diet (*E. coli* OP50 + 40 mM glucose) (lower panel). Diet treatment: 24 hours. Scale bar 20 μm. (**H**) Quantification of endogenous ETS-5-GFP expression in the BAG neurons of animals fed standard diet (*E. coli* OP50, gray bar) and high glucose diet (*E. coli* OP50 + 40 mM glucose, yellow bar), displayed as CTCF values as % of OP50 value. Diet treatment: 24 hours. Data presented as $\bar{x}$ + SEM, *n* shown in bar. Significance assessed by unpaired *t* test. For all data: ns, not significant ($p > 0.05$), * = $p \leq 0.05$, ** = $p \leq 0.01$, *** = $p \leq 0.001$. The underlying numerical data can be found in S1 Data. CTCF, calculated total fluorescence; GFP, green fluorescent protein; RNA interference.

To determine whether ETS-5 directly controls INS-1 expression through promoter binding, we used chromatin immunoprecipitation followed by quantitative PCR (ChIP-qPCR) to measure ETS-5 enrichment at the *ins-1* promoter ETS sites (Fig 3D). To identify endogenous ETS-5 interactions, we used CRISPR-Cas9 genome engineering to tag the ETS-5 locus with DNA encoding GFP (ETS-5-GFP) (S3B Fig). The endogenously tagged ETS-5 protein shows an expected nuclear expression pattern, is the correct size, and functions as the wild-type protein in foraging assays (S3C–S3E Fig). Our ChIP-qPCR analysis revealed that ETS-5-GFP was enriched at the *ins-1* promoter sequence that contains the putative ETS sites, as well as the strongly ETS-5-regulated *flp-17* promoter [37], but not an *ins-1* promoter region devoid of ETS sites (Fig 3D). These data show that ETS-5 directly binds the *ins-1* promoter at the ETS sites. To determine which of the 2 ETS sites were important for ETS-5 regulation of *ins-1* expression, we mutagenized the putative ETS-binding sites in the *ins-1p*::*NLS-GFP* plasmid (S3F Fig). We found that both ETS sites are required for full *ins-1* promoter activity in the

BAG neurons (Fig 3E and 3F). Lacking either ETS site, the *ins-1* promoter activity in the BAG neurons is nearly abolished, whereas the *ins-1* promoter continues to drive expression in the other *ins-1* positive neurons (Fig 3E and 3F), similar to the *ets-5* mutant analysis (Fig 3B and 3C).

## Excess dietary glucose reduces ETS-5 levels in the BAG neurons

As our data implicate ETS-5 as a major regulator of *ins-1* expression in the BAG neurons, we next determined if ETS-5 itself is affected by excess dietary glucose. Using the endogenously tagged ETS-5-GFP strain, we observed that worms fed on 40 mM glucose for 24 hours had significantly decreased ETS-5-GFP levels in the BAG neurons (approximately 80% cf. OP50, Fig 3G and 3H). As both ETS-5 and INS-1 are down-regulated in animals fed a high glucose diet, we asked whether BAG-expressed genes are generally down-regulated under excess glucose conditions, or whether our observations revealed a specific *ins-1* regulatory response to excess glucose. To examine this, we analyzed the expression of 5 other BAG-expressed reporter genes after feeding 40 mM glucose for 24 hours. The reporter genes assayed were for the neuropeptides FLP-13, FLP-17, FLP-19, the transcription factor EGL-13, which are all regulated by ETS-5 [36,51], and the guanylyl cyclase GCY-33, which is only weakly regulated by ETS-5 [36]. We found that under excess dietary glucose diet, *flp-17* expression was mildly reduced (approximately 90% cf. OP50, S4A and S4B Fig) and *flp-19* expression was strongly reduced, and to a similar extent as observed for *ins-1* (approximately 45% cf. OP50, S4A and S4B Fig). The other BAG-expressed reporter genes *flp-13*, *egl-13*, and *gcy-33* were unchanged between OP50 and high glucose diet (S4A and S4B Fig). Our data show that not all ETS-5 regulated genes are affected by excess glucose, and that BAG-expressed genes are not generally down-regulated in response to excess dietary glucose. Rather, our data suggest there is a specific subset of neuropeptide signals, including *ins-1*, *flp-19*, and *flp-17* that are specifically down-regulated in the BAG neurons when animals are fed excess glucose.

## INS-1 controls intestinal fat levels and foraging behavior

The BAG neurons modulate intestinal fat levels and foraging behavior to maintain metabolic homeostasis. We previously showed that ETS-5 acts from the BAGs to reduce fat stores [49], whereas others have shown that the BAG neurons, using the neuropeptide FLP-17, act to promote fat storage [50]. A possible explanation for these opposing BAG functions in lipid metabolism is that the BAG neurons integrate multiple sensory cues, with differentially responsive expression programs for each cue, which adjusts the signaling outputs accordingly. To determine whether INS-1 plays a role in regulating intestinal fat levels, we measured fat levels by Oil-Red O (ORO) staining and a fluorometric triglyceride (TAG) assay in two independent *ins-1* mutant strains (Figs 4A, 4B and S5A). We found that *ins-1* mutant animals stored over 30% more intestinal fat than wild-type animals in the Oil-Red O analysis, with no significant difference observed between the *ins-1* mutant alleles (Fig 4A and 4B), and 75% to 100% more fat in the TAG analysis (S5A Fig). Changes in fat storage are associated with alterations to animal feeding strategies, with increased fat storage leading to decreased food-seeking behavior. This is regulated through the SREB-SCD pathway, an acetyl-coA carboxylase, and multiple nuclear hormone receptors [49,52]. To determine whether the increased intestinal fat storage observed in the *ins-1* mutant affects these strategies, we examined foraging behavior of the two *ins-1* mutant strains using an exploration assay (S5B Fig). Both *ins-1* mutant strains displayed significantly decreased exploration behavior (Fig 4C). However, there was a small but significant difference in exploration behavior between the two *ins-1* alleles that could be attributed to the different genetic backgrounds that slightly affects this behavior, independent of fat storage.

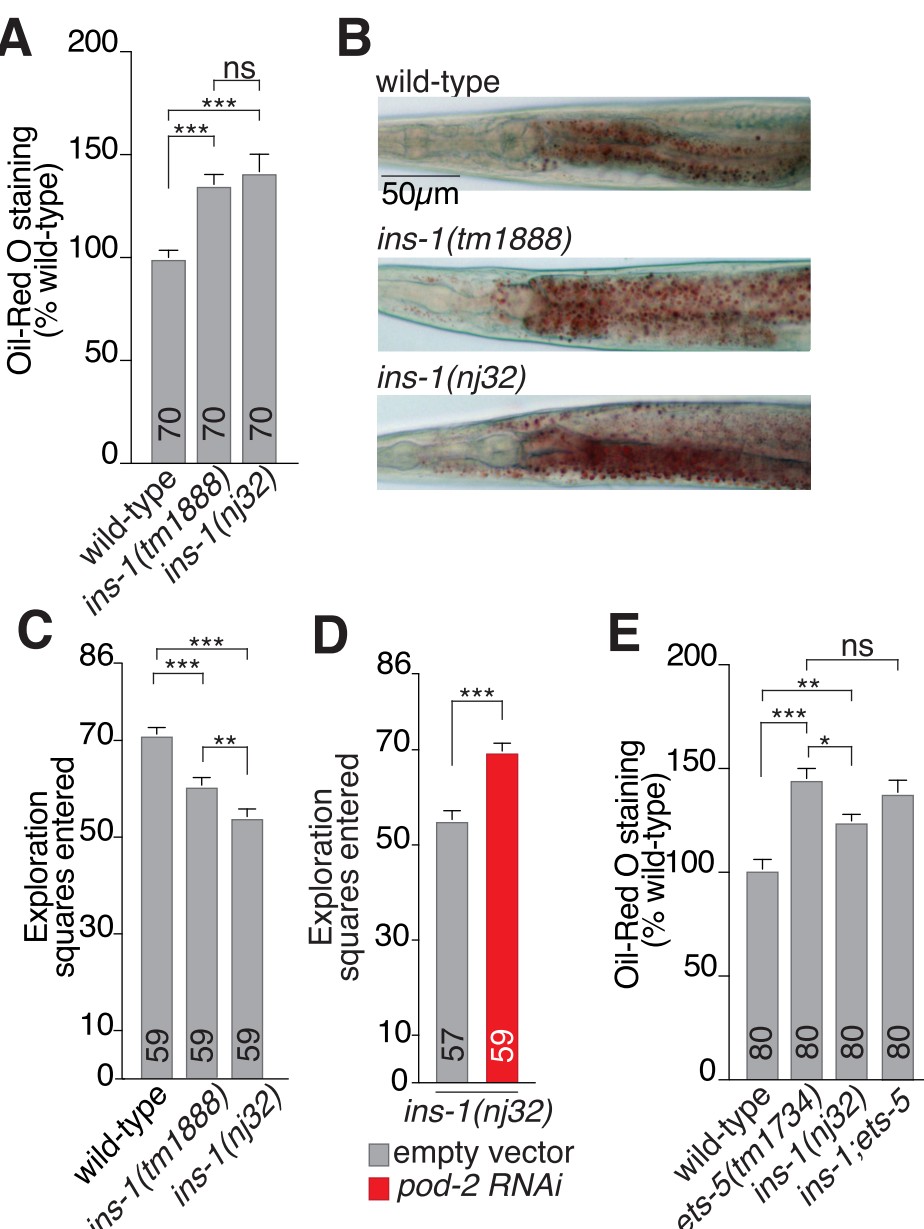

**Fig 4. INS-1 controls intestinal fat levels. (A)** Quantification of ORO staining of wild-type, *ins-1(tm1888)* and *ins-1 (nj32)* animals. Data presented as ORO intensity as % wild-type, $\bar{x}$ + SEM, *n* shown within bar. Significance assessed by 1-way ANOVA (Dunnet's correction). **(B)** Representative ORO images of wild-type, *ins-1(tm1888)* and *ins-1(nj32)* animals. Scale bar 50 μm. **(C)** Exploration assay of wild-type, *ins-1(tm1888)* and *ins-1(nj32)* animals. Data presented as $\bar{x}$ + SEM, *n* shown in bar. Statistical significance assessed by 1-way ANOVA with Tukey's correction for multiple comparisons. **(D)** Exploration assay of *ins-1(nj32)* animals on RNAi plates, treated with either empty vector control (gray) or RNAi targeted to *pod-2* (red). Data presented as $\bar{x}$ + SEM, *n* shown in bar. Statistical significance assessed by unpaired *t* test. **(E)** Double mutant analysis for *ets-5(tm1734)* and *ins-1(nj32)*. ORO quantification of *ins-1(nj32); ets-5 (tm1734)* double mutant compared to each single mutant. Data presented as $\bar{x}$ + SEM, *n* shown within bar. Significance assessed by 1-way ANOVA (Tukey's correction). For all data: ns, not significant ($p > 0.05$), * = $p \leq 0.05$, ** = $p \leq 0.01$, *** = $p \leq 0.001$. The underlying numerical data can be found in S1 Data. ORO, Oil-Red O; RNAi, RNA interference.

The decreased exploration behavior in the *ins-1(nj32)* mutant was dependent on intestinal fat storage, as RNA interference (RNAi) knockdown of a key enzyme in fat synthesis, the acetyl-CoA carboxylase POD-2, rescued the foraging phenotype (Fig 4D). Reduced exploration behavior can result from animals increasing their dwelling behavior, where animals feed in a restricted area, or by an increase in the time spent in quiescence, a sleep-like state where the animals do not feed. To determine which of these behaviors could be leading to the decreased exploration phenotype in *ins-1(nj32)* mutants, we measured the behavior of individual worms as quiescent, dwelling or roaming (an active food search behavior) over a 30-second time period. We found that *ins-1(nj32)* mutants were significantly more likely to be quiescent and less likely to be roaming than wild-type animals (S5C Fig). Together, our data indicate that *ins-1* mutants store more fat, which leads to increased satiety compared to wild-type animals grown on the same food source. Loss of *ets-5* also leads to increased fat storage [49], and we have shown that ETS-5 regulates *ins-1* expression in the BAG neurons. To test whether ETS-5 and INS-1 act in the same pathway to regulate intestinal fat storage, we tested the intestinal fat levels of animals lacking both *ins-1* and *ets-5*. We found that *ins-1; ets-5* compound mutants stored the same level of fat as *ets-5* single mutants, indicating *ins-1* and *ets-5* regulate intestinal fat levels through the same genetic pathway (Figs 4E and S5D). Intriguingly, complete loss of the BAG neurons, through genetic-induced ablation, reduces intestinal fat stores [50], the opposite phenotype of *ets-5* and *ins-1* mutants. We confirmed that BAG ablation significantly reduced intestinal fat storage levels (approximately 50% cf. wild-type) and that removing *ins-1* in the BAG ablation strain did not lead to an increase in fat (S5E Fig). These findings demonstrate that complete loss of the BAG neurons differentially affects intestinal fat levels compared to reducing specific signaling components within the BAG neurons such as INS-1.

INS-1 may act from any or all INS-1 positive neurons to control intestinal fat levels. To determine whether BAG-expressed INS-1 specifically controls intestinal fat levels, we restored *ins-1* expression in the *ins-1(nj32)* mutant using distinct promoters to drive *ins-1* cDNA. The promoters used were (i) the wild-type *ins-1* promoter; (ii) the *ins-1* promoter with both ETS sites mutated; this promoter shows wild-type expression in all *ins-1* neurons except the BAG neurons; (iii) the *flp-17* promoter, which drives expression specifically in the BAG neurons [37]; and (iv) the *ttx-3* promoter, which drives expression specifically in the AIY neurons [53] (Fig 5A). We found that *ins-1* cDNA driven by the wild-type *ins-1* promoter could fully rescue the *ins-1(nj32)* mutant increased fat phenotype (Figs 5B and S5F) and exploration defect (Fig 5C). Whereas resupplying *ins-1* cDNA under the control of the mutated *ins-1* promoter (*ins-1pΔΔETS*) did not rescue fat levels (Figs 5B and S5F) or the exploration defect in *ins-1(nj32)* animals (Fig 5C). It is important to note that the *ins-1* promoter containing mutated ETS-binding sites was active in non-BAG *ins-1* expressing neurons, such as the ASH and AIA neurons (Fig 3E). This means that *ins-1* is still expressed in these other INS-1-positive neurons in the *ins-1pΔΔETS::ins-1cDNA* transgenic lines, yet fails to rescue the *ins-1(nj32)* fat and exploration phenotypes. This suggests that INS-1 acts specifically from the BAG neurons to regulate intestinal fat levels. To corroborate this assertion, we used the BAG-specific *flp-17* promoter to test whether *ins-1* derived specifically from the BAG neurons can control intestinal fat levels and exploration behavior. Transgenic *ins-1* expression controlled by the *flp-17* promoter significantly rescued the *ins-1(nj32)* mutant fat and exploration phenotypes (Figs 5D, 5E, and S5F). In contrast, AIY-specific *ins-1* expression controlled by the *ttx-3* promoter did not rescue the *ins-1(nj32)* mutant fat storage or exploration phenotypes (Figs 5F, 5G and S5F).

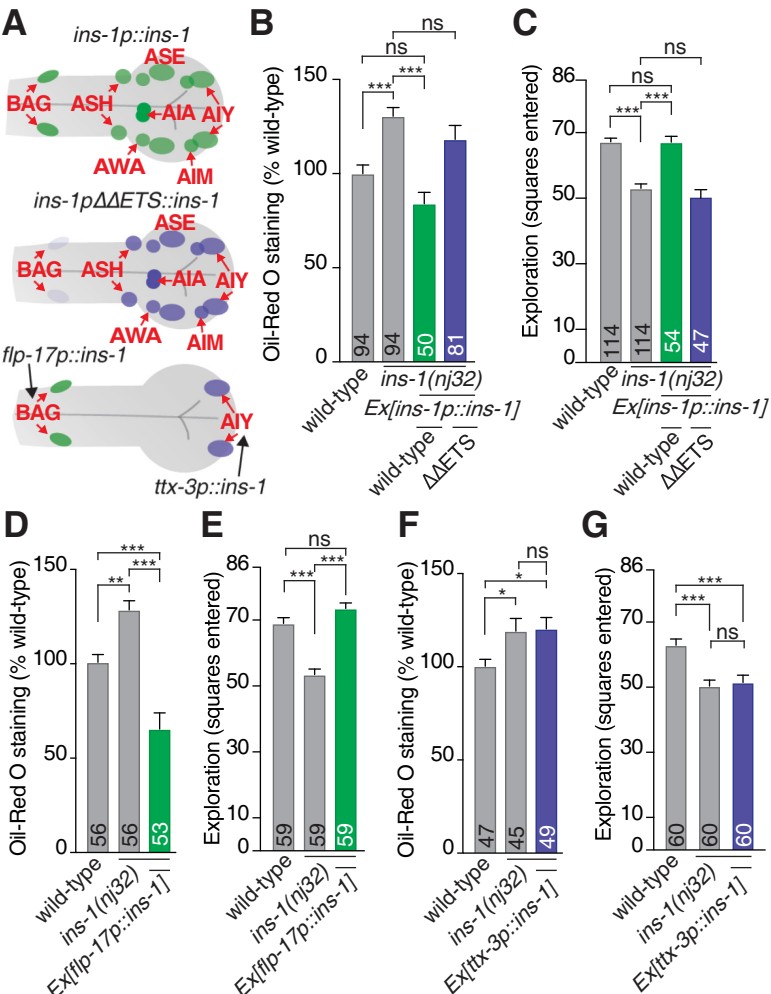

**Fig 5. INS-1 regulates intestinal fat levels specifically from the BAG neurons. (A)** Schematic of expression pattern observed for wild-type *ins-1* promoter (upper panel) and *ins-1* promoter with both ETS site mutated (middle panel), the BAG-specific *flp-17* promoter (green neurons, lower panel), and AIY-specific *ttx-3* promoter (blue neurons, lower panel). **(B)** ORO quantification of *ins-1(nj32)* rescued with wild-type *Ex[ins-1p::ins-1cDNA]* (green bar), or mutant *Ex[ins-1ΔΔETSp::ins-1cDNA]* (blue bar), compared to *ins-1(nj32)* and wild-type. **(C)** Exploration assay of *ins-1(nj32)* rescued with wild-type *Ex[ins-1p::ins-1cDNA]* (green bar), or mutant *Ex[ins-1ΔΔETSp::ins-1cDNA]* (blue bar), compared to *ins-1(nj32)* and wild-type. **(D)** Quantification of ORO staining of *ins-1(nj32)*; *Ex[flp-17p::ins-1cDNA]* (BAG-specific, green bar) compared to *ins-1(nj32)* and wild-type. **(E)** Exploration assay of *ins-1(nj32)*; *Ex[flp-17p::ins-1cDNA]* (BAG-specific, green bar) compared to *ins-1(nj32)* and wild-type. **(F)** Quantification of ORO staining of *ins-1(nj32)*; *Ex[ttx-3p::ins-1cDNA]* (AIY-specific, blue bar) compared to *ins-1(nj32)* and wild-type. **(G)** Exploration assay of *ins-1(nj32)*; *Ex[ttx-3p::ins-1cDNA]* (AIY-specific, blue bar) compared to *ins-1(nj32)* and wild-type. All data presented as $\bar{x}$ + SEM, *n* shown in bar. Significance assessed by 1-way ANOVA (Tukey's correction). ns, not significant ($p > 0.05$), * = $p \leq 0.05$, ** = $p \leq 0.01$, *** = $p \leq 0.001$. The underlying numerical data can be found in S1 Data. ORO, Oil-Red O.

## BAG-INS-1 agonizes canonical insulin signaling in the intestine

*C. elegans* has a single insulin-like receptor with six isoforms called DAF-2 [6,16]. INS-1 can inhibit or activate DAF-2 depending on the developmental, behavioral, or learning context [12,29,30]. For example, INS-1 antagonizes DAF-2 activity in dauer formation, yet agonizes DAF-2 activity in salt-learning behavior [12,30]. In the context of fat regulation, *ins-1* and *daf-2* mutant animals have a similar phenotype—both have increased fat levels [6]; therefore, we

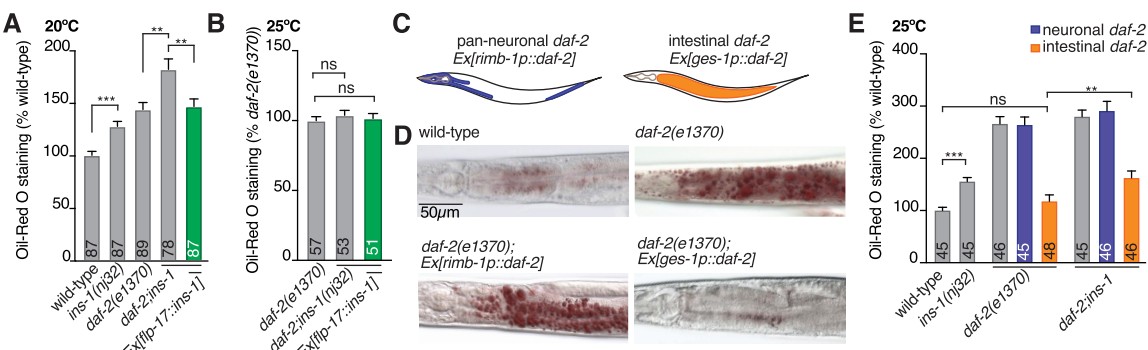

**Fig 6. INS-1 acts from the BAG neurons to control intestinal DAF-2 activity. (A)** ORO quantification of wild-type, *ins-1(nj32)* and *daf-2(e1370)* single mutants and the *daf-2(e1370); ins-1(nj32)* double mutant in the presence or absence of *Ex[flp-17p::ins-1cDNA]*. Assayed at *daf-2(e1370)* semi-permissive temperature of 20°C. **(B)** ORO quantification of *daf-2(e1370)* single mutants and the *daf-2 (e1370); ins-1(nj32)* double mutant in the presence or absence of *Ex[flp-17p::ins-1cDNA]*. Shifted to the *daf-2(e1370)* restrictive temperature 25°C for 24 hours prior to assaying. **(C)** Schematic diagrams showing where *daf-2* is expressed under the *rimb-1* (pan-neuronal) and *ges-1* (intestinal) promoters. **(D)** Representative ORO images of wild-type, *daf-2(e1370)*, *daf-2(e1370); Ex[rimb-1p::daf-2a]* (neuronal *daf-2a* expression) and *daf-2(e1370); Ex[ges-1p::daf-2a]* (intestinal *daf-2a* expression) animal shifted to 25°C for 24 hours prior to staining. Scale bar 50 μm. **(E)** ORO quantification of wild-type, *ins-1(nj32)* and *daf-2(e1370)* single mutants and the *daf-2(e1370); ins-1 (nj32)* double mutant with neuronal (blue bars) and intestinal (orange bars) expression of *daf-2a* cDNA. All data presented as $\bar{x}$ + SEM, *n* shown in bar. Significance assessed by 1-way ANOVA (Tukey's correction). ns, not significant ($p > 0.05$), * = $p \leq 0.05$, ** = $p \leq 0.01$, *** = $p \leq 0.001$. The underlying numerical data can be found in S1 Data. ORO, Oil-Red O.

could expect that INS-1 agonizes DAF-2 activity. However, DAF-2 is more active on a high glucose diet [27]. In this context, we would predict that INS-1 would antagonize DAF-2 activity, as glucose reduces *ins-1* expression, DAF-2 would become more active. To determine the relationship between BAG-derived INS-1 signaling and DAF-2, we first measured fat levels in *daf-2(e1370); ins-1(nj32)* compound mutant animals. The *daf-2(e1370)* allele is temperature sensitive; 20°C is a semipermissive temperature that enables partial DAF-2 function [54]. At 20°C growth, we observed that the *daf-2(e1370); ins-1(nj32)* compound mutant exhibits additive fat levels (180% cf. wild-type) compared to either single mutant alone (*daf-2(e1370)* = 140% cf. wild-type, *ins-1(nj32)* = 130% cf. wild-type) (Fig 6A). BAG-specific *ins-1* expression rescued this additive effect on fat storage to *daf-2* single mutant levels (Fig 6A). We then measured fat levels of animals transferred to 25°C, the temperature that restricts DAF-2 function, 24 hours prior to fat staining. At the restrictive temperature, the *daf-2(e1370); ins-1(nj32)* compound mutant and *daf-2(e1370); ins-1(nj32); Ex[flp-17p::ins-1]* fat levels were identical to the *daf-2(e1370)* single mutant (Fig 6B). As loss of *ins-1* in the *daf-2(e1370)* mutant leads to an increase in fat storage at 20°C, these data suggest that BAG-expressed INS-1 activates DAF-2, and when INS-1 is absent DAF-2 activity is decreased.

To determine where INS-1 regulates the DAF-2 insulin-like receptor, we performed ORO experiments on *daf-2(e1370)* animals expressing *daf-2a* cDNA pan-neuronally using the *rimb-1* promoter or in the intestine using the *ges-1* promoter (Fig 6C–6E). We found that pan-neuronal *daf-2a* cDNA expression did not rescue the *daf-2(e1370)* fat storage phenotype, whereas intestinal *daf-2a* cDNA expression fully rescued *daf-2(e1370)* to wild-type fat levels (Fig 6D and 6E). Further, we found that the *daf-2(e1370); ins-1(nj32)* compound mutant had higher fat levels than *daf-2(e1370)* when *daf-2a* is resupplied to the intestine, indicating that INS-1 agonizes DAF-2 in the intestine and not the nervous system to control intestinal fat storage (Fig 6E). A major downstream target of DAF-2 signaling is the FOXO transcription factor DAF-16 [55,56]. Active DAF-2 signaling inhibits DAF-16 localisation to the nucleus, and reduced DAF-2 activity leads to DAF-16 translocation to the nucleus, where it regulates gene

expression to alter the metabolic profile of *C. elegans* (reviewed in [57]). We used RNAi to knock down *daf-16* levels in wild-type, *ins-1(nj32)*, *daf-2(e1370)*, and *daf-2(e1370); ins-1(nj32)* mutants. In agreement with previous findings, DAF-16 was essential for the increased fat phenotype of the *daf-2(e1370)* mutant ([55], Fig 7A). Consistent with our observation that BAG-derived INS-1 is a DAF-2 agonist, we found that DAF-16 was required for the increased fat storage of *ins-1(nj32)* mutants (Fig 7A).

These above observations are, however, inconsistent with the current understanding that insulin signaling is activated upon high glucose feeding. In the context of life span, a high glucose (111 mM glucose) diet did not further reduce the short life span of *daf-16* mutants [26]. However, another study that fed worms with a diet of heat-killed bacteria that were fed 22 mM glucose showed that DAF-16::GFP localisation did not change, but total fluorescence

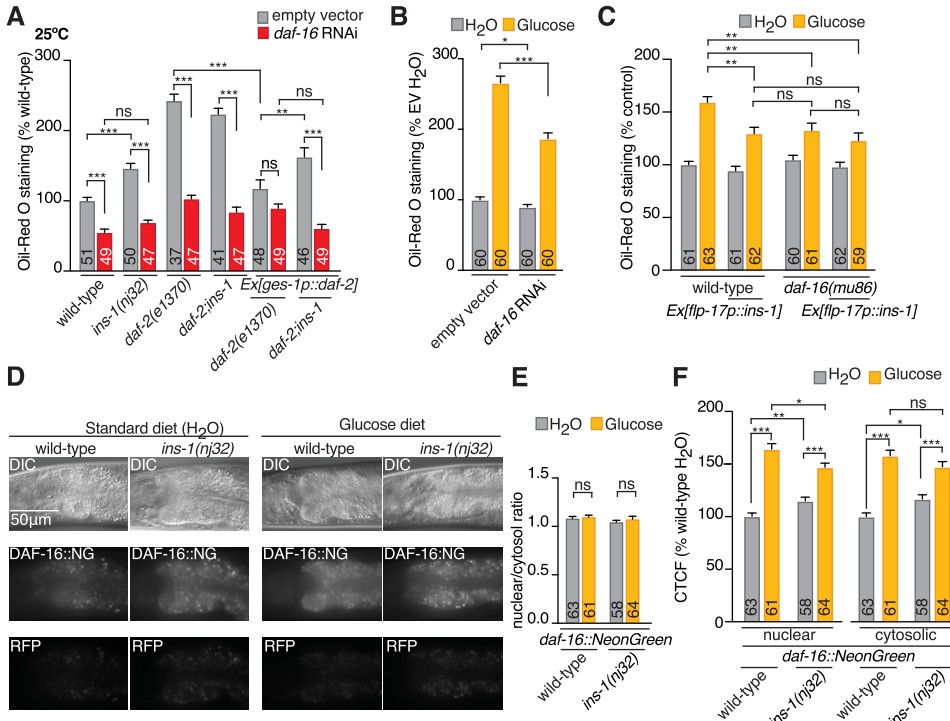

**Fig 7. INS-1 action is DAF-16 dependent and modulates intestinal DAF-16 expression. (A)** ORO quantification of wild-type, *ins-1(nj32)* and *daf-2(e1370)* single mutants, *daf-2(e1370); ins-1(nj32)* double mutants, *daf-1(e1370);Ex[ges-1p::daf-2a]* lines and *daf-2(e1370); ins-1(nj32); Ex[ges-1p::daf-2a]* lines with (red bars) and without (gray bars) *daf-16* RNAi treatment. *daf-16* RNAi from L1 stage, animals shifted to restrictive temperature 25˚C for 24 hours prior to assaying. **(B)** ORO quantification of wild-type animals treated with either EV control or RNAi targeted to *daf-16* from L1 stage, shifted to HT115 + RNAi + H₂O (H₂O) or HT115 + RNAi + 40 mM glucose (Glucose) plates, containing the matching RNAi treatment 24 hours prior to staining. **(C)** ORO quantification of wild-type and *daf-16(mu86)* mutants with and without BAG-expressed *ins-1cDNA* (Ex*[flp-17p::ins-1]*) shifted to OP50 + H₂O (H₂O) or OP50 + 40 mM glucose (Glucose) plates 24 hours prior to staining. **(D)** Representative micrographs of DIC, DAF-16::NeonGreen (DAF-16::NG), and autofluorescence (RFP) in wild-type and *ins-1(nj32)* mutants shifted to OP50 + H₂O (H₂O) or OP50 + 40 mM glucose (Glucose) plates 24 hours prior to imaging. **(E)** Calculation of nuclear:cytosol DAF-16::NG levels in wild-type and *ins-1(nj32)* mutants shifted to OP50 + H₂O (H₂O) or OP50 + 40 mM glucose (Glucose) plates 24 hours prior to imaging. **(F)** Quantification of normalized cellular fluorescence levels (CTCF) of DAF-16::NG in the nuclei and cytosol of wild-type and *ins-1(nj32)* mutants shifted to OP50 + H₂O (H₂O) or OP50 + 40 mM glucose (Glucose) plates 24 hours prior to imaging. A–C and E and F significance assessed by mixed-effects model (2-way ANOVA). All data presented as $\bar{x}$ + SEM, *n* shown in bar. A–C and E and F significance assessed by mixed-effects model (2-way ANOVA). ns, not significant ($p > 0.05$), * = $p \leq 0.05$, ** = $p \leq 0.01$, *** = $p \leq 0.001$. The underlying numerical data can be found in S1 Data. CTCF, calculated total fluorescence; EV, empty vector; ORO, Oil-Red O; RFP, red fluorescent protein; RNAi, RNA interference.

decreased, while *daf-16* mRNA increased by 20% [58]. How DAF-2 and DAF-16 act is clearly impacted by the context, timing and downstream readout of high glucose feeding, and is likely highly nuanced. To further dissect how DAF-16 acts in the BAG-INS-1 pathway, we tested the role of DAF-16 in fat storage in response to short-term excess dietary glucose. First, we used RNAi to knock down *daf-16* from the synchronized L1 stage, then transferred animals at the L4 stage to control or 40 mM glucose + OP50 plates. We then measured the level of fat deposition by ORO staining. As expected, we found that 24-hour exposure to excess glucose increased fat levels in both empty vector control and *daf-16* RNAi animals (Fig 7B). However, we found that knockdown of *daf-16* resulted in significantly less fat deposition on the excess glucose diet compared to control animals (Fig 7B). Using the *daf-16(mu86)* null mutant produced comparable results to the *daf-16* RNAi, demonstrating that the *daf-16* RNAi knockdown worked efficiently, and showing that DAF-16 function is required to accumulate fat when animals are fed excess glucose (Fig 7C). If BAG-derived INS-1 activates DAF-2 and thus inhibits DAF-16, then we predict that overexpressing INS-1 from the BAG neurons would produce a similar effect as loss of *daf-16* on intestinal fat levels. This is because DAF-2 will be activated, and therefore DAF-16 activity would be repressed. Similar to our *daf-16* loss-of-function data, we found that overexpressing *ins-1* from the BAG neurons reduced intestinal fat storage when fed excess glucose compared to wild-type worms (Fig 7C). We also found that overexpressing *ins-1* from the BAG neurons did not further reduce the fat levels of *daf-16 (mu86)* animals when fed excess glucose, indicating that *ins-1* and *daf-16* act in the same genetic pathway to regulate intestinal fat levels in response to excess glucose (Fig 7C).

Previous studies have typically used much higher levels of glucose throughout development than the conditions used in this study. Therefore, the nuances in how insulin-like signaling functions in acute response to excess glucose compared to a major metabolic challenge are not clear. Here, we found that DAF-16 is required for generating approximately 34% of the fat accumulated over 24 hours on 40 mM glucose (Fig 7C). This means that DAF-16 is not required for accumulating most (approximately 66%) of the fat increase observed when animals are fed 40 mM glucose for 24 hours. We therefore hypothesize that the BAG-INS-1 signaling pathway acts on a specific intestinal metabolic pathway that requires DAF-16, compared to the majority of glucose-induced metabolic changes that are DAF-16 independent. Based on our hypothesis, we predict that there would be an increase in nuclear-localized DAF-16 in the *ins-1* mutant, and when animals are fed excess glucose. Rather than examine the traditional DAF-16::GFP translation reporter, which is overexpressed, we used a recently available CRISPR-tagged *daf-16::NeonGreen* allele (*daf-16::NG*) [59]. To determine whether *daf-16::NG* responds to a canonical stressor, we performed a heat-shock control experiment. DAF-16 accumulates in the nucleus after heat shock at 37˚C [60,61], therefore we placed *daf-16::NG* animals at 37˚C for 45 minutes, then measured the GFP fluorescence in the cytosol and nucleus of the first 2 intestinal cells (S6A–S6C Fig). As expected, the nuclear/cytosolic ratio of DAF-16::NG after heat shock significantly increased (S6B Fig). We also compared the nuclear and cytosolic DAF-16::NG levels between control and heat-shocked animals. We found that while the nuclear DAF-16::NG levels increased upon heat shock, there was not a comparable decrease in cytosolic DAF-16::NG (S6C Fig), suggesting that DAF-16 expression and/or stability increases upon heat shock.

We measured endogenous DAF-16::NG expression in wild-type and *ins-1(nj32)* mutant backgrounds, under standard OP50 and excess glucose diets (Fig 7D–7F). We found that the nuclear/cytosolic ratio of DAF-16::NG was unchanged in the *ins-1(nj32)* mutant or on excess glucose (Fig 7E). However, the level of DAF-16::NG in both the cytosol and nucleus is significantly increased (approximately 115% cf. wild-type) in the *ins-1(nj32)* mutant compared to wild-type animals (Fig 7F). We further found that DAF-16::NG levels were strongly increased

(approximately 160% cf. standard diet) in animals fed excess glucose compared to standard diet (Fig 7F). Our data show that *ins-1* expression is down-regulated in the BAG neurons even when the animals are fed nonmetabolizable L-glucose (Fig 2E). Based on these data, we hypothesize that excess dietary L-glucose would also lead to increased intestinal DAF-16 levels. Therefore, we fed animals L-glucose for 24 hours and measured DAF-16::NG nuclear and cytosolic levels in the intestine. Consistent with our hypothesis, we found that L-glucose treatment significantly increased DAF-16::NG levels (approximately 128% cf. standard diet) in both the nucleus and cytosol (S6D and S6E Fig). These data show that endogenous DAF-16 expression increases on the glucose diet corroborate our genetic observations that INS-1 modulates the DAF-2/DAF-16 pathway in response to glucose.

DAF-16 has been shown to affect the expression, either directly or indirectly, of thousands of genes (consolidated in [62]). A well-characterized downstream target of insulin-like signaling is the mitochondrial superoxide dismutase SOD-3. SOD-3 expression increases in *daf-2* mutants in a *daf-16*-dependent manner [63,64]. We therefore tested whether the observed increase in DAF-16::NG levels on excess glucose and in *ins-1(nj32)* mutants corresponded to changes in SOD-3 expression. We found that SOD-3 expression significantly increased in the first 2 intestinal cells of animals fed excess glucose (S6F and S6G Fig). This is consistent with the idea that animals grown on a high glucose diet suffer oxidative damage to their mitochondria [34,35]. Loss of INS-1 does not appear to have the same effect as excess glucose, as loss of *ins-1* did not alter SOD-3::GFP levels in the intestine (S6F and S6G Fig). We therefore hypothesize that the increase in overall DAF-16 levels caused by reduced BAG-derived INS-1 could lead to specific transcriptional changes that alter intestinal metabolism, without increasing oxidative stress, in response to excess glucose.

## Discussion

Here, we have revealed a mechanism that drives the specific spatial and temporal expression of a neuronal insulin-like peptide that is important for controlling systemic metabolism and foraging behavior. We have shown that the *C. elegans* insulin ortholog, INS-1, acts from the BAG sensory neurons to agonize DAF-2 activity in the intestine and repress DAF-16 levels. INS-1 expression in the BAG neurons is reduced in response to excess glucose, which leads to decreased DAF-2 activity and increased DAF-16 levels that result in increased fat storage (see model, Fig 8). BAG-INS-1 expression responds to excess dietary glucose in developed animals. Therefore, the adjustment in INS-1 levels in response to glucose would not alter the developmental wiring of the worm but is a mechanism to modulate neuronal signaling postdevelopmentally to maintain metabolic homeostasis.

We found that INS-1 regulates intestinal fat specifically from the BAG neurons, as INS-1 derived from other neurons did not regulate intestinal fat levels. This demonstrates a BAG-specific INS-1 function that is different from the functions in learning, memory, and development that INS-1 performs from other neurons [28–31]. How INS-1 derived specifically from the BAG neurons is able to convey a different signal from INS-1 produced in other neurons is a particularly intriguing question. The BAG neurons spatial location in the nervous system may contribute to the specificity of BAG-INS-1 action. However, it is possible that specific structural properties, such as posttranslational modifications, could also play a role in differentiating BAG-expressed INS-1 from non-BAG-expressed INS-1. Such structural specificity would enable BAG-expressed INS-1 to perform its function in controlling fat metabolism independently of, or without interfering with, other INS-1 functions. The data presented here demonstrate that the same signaling peptide, expressed from different neurons, can drastically affect the output of that signal. This observation highlights the complexity of neuropeptide

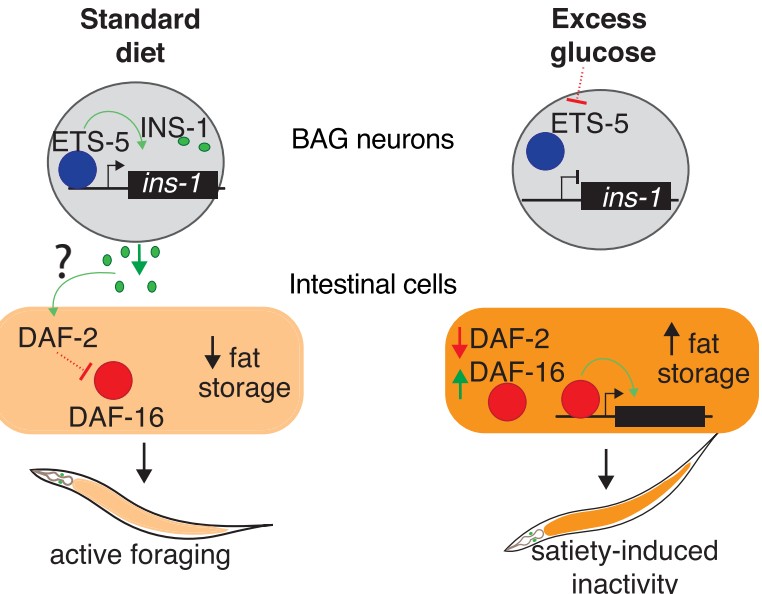

**Fig 8. Model of the ETS-5/INS-1 regulatory module.** Under standard food conditions, ETS-5 directly promotes *ins-1* expression in the BAG neurons. INS-1 acts from the BAG neurons to reduce fat levels in the intestine by activating DAF-2, which inhibits DAF-16 activity. Reduced fat stores lead to increased foraging behavior. Worms that experience excess dietary glucose down-regulate ETS-5 and INS-1 levels in the BAG neurons. This suppression of ETS-5/INS-1 leads to decreased intestinal DAF-2 activity, and increased intestinal DAF-16 levels. DAF-16 then alters intestinal gene expression to promote fat storage and produces a satiety-induced reduction in foraging activity.

signaling and has implications for our understanding of neuropeptide action in any type of nervous system [65]. Our observations show that BAG-derived INS-1 targets intestinal DAF-2. However, insulin-like signaling is complex, integrates 40 insulin-like peptide signals and forms intricate regulatory networks [66]. It is possible that BAG-INS-1 alters the activity and levels of other insulin-like peptides within the nervous system to alter intestinal DAF-2 activity. Whether BAG-INS-1 directly acts on intestinal DAF-2, or modulates the activity of other insulin-like peptides that affect intestinal DAF-2 activity, will require further analysis.

Our data demonstrate that ETS-5 regulates the spatial specificity of INS-1, by controlling *ins-1* expression in the BAG neurons. We also found that ETS-5, and two additional ETS-5-regulated neuropeptide genes, *flp-19* and *flp-17*, were down-regulated in response to excess glucose. Our finding that not all ETS-5-regulated genes are down-regulated in response to excess glucose suggests targeted regulation of a subset of ETS-5-regulated genes by glucose. This may be achieved by each gene having a different sensitivity to reduced ETS-5 levels, such as altering the number of ETS sites in the promoter, the binding affinities of ETS-5 at those promoter sites, different cofactors that regulate these genes alongside ETS-5, and the epigenetic context of each gene. Our results support a mechanism where, when there is excess dietary glucose, nuclear ETS-5 levels are reduced, and a specific module of ETS-5-regulated neuropeptide genes is inhibited, which alters the signature of BAG neuropeptide signaling.

It is clear that the BAG neurons act in a complex, context-dependent manner to integrate external and internal signals to appropriately modulate physiology. Our finding that ETS-5 is down-regulated under high nutrient conditions has broader implications for the diverse BAG-directed behaviors, and provides insight into how various environmental inputs can be integrated by the nervous system to elicit the appropriate response. The BAG neurons play parallel roles in oxygen ($O_2$) sensing, acute carbon dioxide ($CO_2$) responses, egg laying, and

modulation of other sensory neurons [50,67–70]. Complete BAG ablation, or reducing the BAG neurons ability to sense low $O_2$, leads to reduced fat levels (S5D Fig) [50], whereas removing the ETS-5-mediated transcriptional program within the BAGs leads to increased fat levels [49]. Absence of the BAGs abolishes all neuropeptide/neurotransmission signals, and may cause developmental rewiring. However, we also speculate that the opposing BAG functions are vital for their role in maintaining metabolic homeostasis. In essence, the BAGs act to integrate sensory information about the outside environment (changes to $O_2$) with the inside environment (fluctuations in nutrient availability). They would then adjust their signaling accordingly and convey the appropriate metabolic and behavioral outputs to maintain physiological homeostasis.

The role of insulin-like signaling in response to excess glucose is also complex and is clearly not an ON/OFF switch, which is to be expected for such an important mechanism controlling physiological homeostasis. The majority of previous reports of insulin-like signaling function and its relation to glucose are with respect to life span, and many studies have used an approximately 3 times higher glucose diet (2% or around 110 mM) than was used in this study (0.7% or 40 mM)[26]. This glucose concentration is still high enough to induce life span changes, mitochondrial damage, egg-laying defects, and lead to significant increases in fat storage. Usually, embryos are placed on the high glucose diet and adults are analyzed, which could result in neuronal rewiring and significant changes to how gene expression and metabolic programs are established during development. For example, feeding a high glucose diet over the entire developmental period of the animal, causes severe mitochondrial damage [25,35], which would require systemic changes to metabolic programs and mitochondrial stress responses. The time frame of treatment used in our work is much shorter than previous studies, and we expect that the time is short enough so that animals are able to mitigate the effects of excess glucose by activating the appropriate responses, yet is not so long that metabolic homeostasis has broken down. Our data show that animals still gain a significant amount of fat in the 24 hours on high glucose. The stress-response systems required to mitigate damage from this excess glucose are likely to be activated but not overwhelmed in this time frame, as our data showing a small but significant SOD-3::GFP induction would suggest. One aspect for further study is to take into account that different cell types have different metabolic needs; therefore, we could expect that tissues other than the intestinal cells respond to excess glucose differently in this timeframe. Our readout of BAG-INS-1 function was changes to intestinal fat storage, yet it is possible that BAG-INS-1 also affects the metabolic programs of muscle and neurons—that there is differential metabolic reprogramming by BAG-INS-1 in other cell types is an intriguing possibility.

We have also shown that the *ins-1* transcriptional reporter is down-regulated in the presence on the nonmetabolizable L-glucose, whereas previous phenotypes have not been observed when animals are fed L-glucose—suggesting that downstream metabolic processes that contribute to the previously observed phenotypes are not comparable to the response to free glucose we observed for *ins-1*. Our finding that L-glucose leads to *ins-1* down-regulation raises the intriguing possibility that this regulation can be mediated by non-enzymatic glycation. Protein glycation plays an important role in the pathology of diabetes [71], and because it is non-enzymatic, it can be achieved by both D- and L-glucose. Non-enzymatic glycation lacks sequence motif bias, so targets of this posttranslational modification are difficult to predict, identify, and validate. The identity of the glycation target upstream of *ins-1* regulation could possibly be ETS-5 or it may be another unknown molecule or molecules. Identifying upstream targets will be facilitated by the advancement of new technologies and methods for studying non-enzymatic glycation, and will be an important aspect of future work.

Maintaining metabolic homeostasis is complex, and the response to excess dietary glucose is multifaceted including insulin-like signaling, TGF-β, SBP-1, and nuclear hormone receptors. We propose that the BAG-INS-1 mechanism of action lies as a specific component within the complex insulin-like signaling network. For example, particular DAF-16 isoforms can carry out specific functions [72]. As the fluorescently tagged *daf-16* allele used in our analysis labels all of the 11 known DAF-16 isoforms, we are unable to determine whether the levels of all DAF-16 isoforms or just particular isoforms increase when *ins-1* is absent, or in the presence of excess glucose. Deciphering this would be a key next step in understanding the specifics of how BAG-INS-1 acts to modulate intestinal fat levels. BAG-INS-1 action may be involved in modulating a specific type of fat accumulation. Reducing DAF-2 function is usually associated with starvation, where animals store increased amounts of poly-unsaturated fatty acids (PUFAs). A high glucose diet is usually associated with activating DAF-2 function, and increased saturated fatty acid (SFA) accumulation. Therefore, determining whether BAG-INS-1 affects PUFA, MUFA (mono-unsaturated fatty acids), SFA accumulation, or particular classes of lipids would help untangle the BAG-INS-1 mode of action.

## Materials and methods

### *C. elegans* culture

Strains used in this work are listed in S1 Table. *C. elegans* were grown using standard growth conditions on NGM agar at 20˚C on *E. coli* OP50, unless otherwise stated [73]. All mutant strains were backcrossed to N2 a minimum of 3 times, and animals were well fed for at least 2 generations prior to experiments.

### Preparation of glucose and other metabolite-enriched OP50 plates

NGM plates (6 cm/10 mL) were fully covered with 400 μL of a 1 M stock solution prepared in diH$_2$O of D-glucose (D-(+)-glucose; Sigma-Aldrich, Australia), D-sorbitol (Sigma-Aldrich), L-glucose (L-(-)-glucose; Sigma-Aldrich), to reach the desired concentration of 40 mM, or with 400 μl diH$_2$O as control. For Palmitic acid 100 mM working stocks were prepared as in [74] and a final concentration of 0.1 mM was used. Plates were allowed to dry for 24 hours then, 400 μL of *E. coli* OP50 was added to cover the entire plate and incubated for 2 nights at room temperature. For heat-killed OP50 assay, OP50 cultures were grown without shaking at 37˚C for 72 hours, then heat-killed at 65˚C for 40 minutes. No bacterial growth was detected after 2 days.

### Glucose and other metabolite treatment

Well-fed mid-stage L4 animals were picked from standard OP50 growth plates and transferred to test plates. Animals were then grown at 20˚C for 24 hours before imaging.

### Oil-Red-O staining

ORO staining was performed essentially as described previously [75]. Detailed methods are included in the S1 Text.

### Total triglyceride quantification

Was performed using the BioVision triglyceride quantification colorimetric/fluorometric kit (BioVision, catalog number:K622), following the protocol described in [76].

## Fluorescence microscopy

For *ins-1p*::*NLS-GFP* and other BAG reporter gene analysis: Animals were anesthetized with 20 mM NaN$_3$ on 5% agarose pads, and images were obtained with an Axio Imager M2 fluorescence microscope, Axiocam 506 mono camera and Zen software (Zeiss, Australia). Fluorescence images were obtained using a 100 X oil objective, and collected as Z-stacks through the worm head with 1 μm step size. For ORO analysis: Images were obtained with a 40 X objective, using mCherry, GFP, and DAPI filters with transmitted light to obtain RGB images. For DAF-16::NG and SOD-3::GFP analysis: Animals were anesthetized with 0.1 ng/mL Levamisole. For DAF-16::NG images were obtained using 64 X oil objective, focused on the nuclei of the first two intestinal cells with both GFP and red fluorescent protein (RFP) images obtained. For SOD-3::GFP analysis images were obtained with 40 X objective.

## Quantification and statistical analysis

All experiments were performed in 3 independent replicates, *n* values are indicated in the bars of corresponding experiments. BAG fluorescence intensity was quantified in FIJI (ImageJ) by tracing the BAG nuclei from DIC images, then changing to the fluorescence (GFP) channel and the integrated density, mean gray, and area were measured. ASH nuclei were traced through the fluorescence channel. For each fluorescent measurement, a background gray value measurement was made in an area within the worm that showed no specific fluorescence (pharynx in most cases) in order to calculate total cell fluorescence (CTCF). ORO staining was quantified in FIJI (ImageJ) by tracing the first 4 intestinal cells proximal to the pharynx. The intensity in these cells was measured in the inverted green channel (where ORO absorbs the light), collecting area, mean gray value, and integrated density data. The pharynx region was used as background measurement for ORO staining, and used to calculate the CTCF. CTCF is calculated as: *Integrated Density–(area * mean gray of background)*. For *DAF-16::NeonGreen* analysis, the background autofluorescence intensity of the RFP channel was used as background gray value for the CTCF calculation. This was to remove bias of increased autofluorescence particles observed in the intestinal cells when animals have increased fat storage. ChIP-qPCR Ct values were converted into % input with the following calculation: Calculated 100% input = Average Ct value (input) − 3.3. Calculated % input = $100 \times 2^{(100\% \text{ input} - \text{Ct (ChIP)})}$. Calculating DAF-16-independent (DAF-16$^i$) fat accumulation of glucose was calculated as: $X^i = \Delta$DAF-16 (glucose-H2O)/$\Delta$WT (glucose-H2O)*100, DAF-16-dependent (DAF-16$^d$) fat accumulation is: DAF-16$^d$ = 100-DAF-16$^i$, using data from Fig 7C. Statistical analysis was performed in GraphPad Prism 7 using mixed-effects analysis with Tukey's Multiple Comparison Test (when both genotype and feeding variables were altered), 1-way analysis of variance (ANOVA) followed by Dunnet's Multiple Comparison Test or Tukey's Multiple Comparison Test (when single variable was compared), or unpaired *t* test (for paired comparisons), or ratio-paired *t* test (ChIP-qPCR), indicated in figure legends. Values are expressed as mean ± SEM, where possible individual data points have been plotted. Differences with a *p*-value < 0.05 were considered significant.

Additional method details are described in the S1 Text.

## Supporting information

**S1 Fig. *ins-1*-reporter expression pattern analysis. (A)** Data accumulated from multiple single-cell RNA-seq experiments by Lorenzo and colleagues [39]. Data show the normalized RNA-seq count of *ins-1* across the identified *ins-1* expressing neurons. **(B)** Representative images of the *ins-1p*::*NLS-GFP* reporter expression pattern in the head. Only neurons whose identity could be clearly and consistently recognized were used for analysis. **(C)** Expression of

*ins-1p*::*NLS-GFP* in a pair of unidentified ventral nerve cord neurons were assessed for response to glucose. Upper panel: representative images, lower panel: quantification. **(D)** Expression of *ins-1p*::*NLS-GFP* in a pair of unidentified tail neurons were assessed for response to glucose. Upper panel: representative images with the most distal intestinal cell circled with a gray dashed line, lower panel quantification. C and D data presented as CTCF % of no glucose, $\bar{x}$ + SEM, *n* shown within bars, ns, not significant ($p > 0.05$), significance assessed by unpaired *t* test. The underlying numerical data can be found in S1 Data. CTCF, calculated total fluorescence; GFP, green fluorescent protein.
(EPS)

**S2 Fig. Excess glucose acts directly on the BAG neurons to down-regulate *ins-1*. (A)** Representative micrographs of *ins-1p*::*NLS-GFP* expression on standard diet compared to glucose diet in genotypes quantified in Fig 2A–2D. **(B)** Representative micrographs of *ins-1p*::*NLS-GFP* expression on standard diet, D-glucose, palmitic acid, L-glucose, and D-sorbitol, quantified in Fig 2E. GFP, green fluorescent protein.
(EPS)

**S3 Fig. Assessing the BAG-specifying transcription factor ETS-5 in regulating *ins-1* expression. (A)** ins-1p::NLS-GFP expression colocalises with a *ets-5p*::*mCherry* transcriptional reporter only in the BAG neurons. **(B)** ETS-5 was C-terminally tagged at the genomic locus using CRISPR/Cas9. **(C)** Endogenously-tagged ETS-5 colocalises in the BAG and ASG neurons with a previously described *ets-5p*::*mCherry* transcriptional reporter. **(D)** Endogenously tagged ETS-5-GFP is the correct protein size (ETS-5 = approximately 23 kDa, GFP = 27 kDa, combined = approximately 50 kDa) when analyzed using SDS-PAGE followed by western blot. The original blot image can be found in S1 Raw Image. **(E)** The endogenously tagged ETS-5::GFP is functional—showing no defect in exploration behavior, which is observed in *ets-5(tm1734)* mutants. *n* values shown in bars, ns, not significant ($p > 0.05$), *** = $p \leq 0.001$. Statistical significance assessed by 1-way ANOVA with Tukey's correction for multiple comparisons. **(F)** ETS motifs identified in the *ins-1* promoter (green). Mutations generated to disrupt the ETS motif sequence are shown in red. The underlying numerical data can be found in S1 Data. GFP, green fluorescent protein.
(EPS)

**S4 Fig. Some BAG-expressed genes are regulated by a high glucose diet. (A)** Representative images of reporter genes *flp-13p*::*GFP*, *flp-17p*::*GFP*, *flp-19p*::*GFP*, *egl-13p*::*GFP*, and *gcy-33p*::*GFP* fed standard diet *E. coli* OP50 +H$_2$O (left panels) and excess glucose diet *E. coli* OP50 + 40 mM glucose (right panels). Diet treatment time: 24 hours. **(B)** Quantification of GFP in the BAG neurons of *flp-13p*::*GFP*, *flp-17p*::*GFP*, *flp-19p*::*GFP*, *egl-13p*::*GFP*, and *gcy-33p*::*GFP* reporter strains fed standard diet (*E. coli* OP50 + H$_2$O, gray bar) and excess glucose diet (*E. coli* OP50 + 40 mM glucose, yellow bar), displayed as CTCF values as % of OP50 value. Diet treatment: 24 hours. Data presented as $\bar{x}$ + SEM, *n* shown in bar. Significance assessed by unpaired *t* test. ns, not significant ($p > 0.05$), * = $p \leq 0.05$, ** = $p \leq 0.01$, *** = $p \leq 0.001$. The underlying numerical data can be found in S1 Data. CTCF, calculated total fluorescence; GFP, green fluorescent protein.
(EPS)

**S5 Fig. INS-1 regulates intestinal fat storage and feeding behavior. (A)** Total triacylglyceride concentration of wild-type, *ins-1(tm1888)* and *ins-1(nj32)* animals. Significance assessed by Welch's *t* test between mutant and wild-type measurements **(B)** Schematic of exploration assay. **(C)** Qualitative behavioral assay showing fraction of wild-type or *ins-1(nj32)* mutant animals displaying quiescent, dwelling, or roaming behavior. Significance assessed by Chi-square.

**(D)** Representative micrographs of ORO staining in wild-type, *ins-1(nj32)*, *ets-5(tm1734)*, and *ins-1(nj32); ets-5(tm1734)* double mutants quantified in Fig 4F. **(E)** ORO quantification of wild-type and *ins-1(nj32)* mutants alone (control) or carrying transgenic caspase constructs that induce BAG ablation. Significance assessed by mixed-effects model (2-way ANOVA), data presented as $\bar{x}$ + SEM, *n* shown in bar. **(F)** Representative micrographs of ORO staining in the *ins-1cDNA* rescue lines quantified in Fig 5B, 5D and 5F. ns, not significant ($p > 0.05$), $^{*}$ = $p \leq 0.05$, $^{**}$ = $p \leq 0.01$, $^{***}$ = $p \leq 0.001$. The underlying numerical data can be found in S1 Data. ORO, Oil-Red O.
(EPS)

**S6 Fig. DAF-16 levels increase in ins-1(nj32) mutants and after D- and L-glucose feeding.** **(A)** Representative micrographs of DIC, DAF-16::NeonGreen (DAF-16::NG), and autofluorescence (RFP) under non-heatshock and HS conditions. **(B)** Quantification of the nuclear/cytosol ratio of normalized fluorescence between HS and non-heatshock (no HS) DAF-16::NeonGreen animals. Treatment 37°C, 45 minutes. **(C)** Quantification of calculated total fluorescence (CTCF) of HS relative to non-heatschock (no HS) in the nucleus and cytosol compartments of DAF-16::NeonGreen animals. **(D)** Representative micrographs of DIC, DAF-16::NeonGreen (DAF-16::NG), and autofluorescence (RFP) in animals fed standard diet (OP50 +H2O), OP50+ 40 mM D-glucose, or OP50+40 mM L-glucose. Treatment time 24 hours. **(E)** Quantification of nuclear and cytosolic CTCF of DAF-16::NeonGreen animals fed standard diet (OP50+H2O), OP50+ 40 mM D-glucose, or OP50+40 mM L-glucose. Treatment time 24 hours. **(F)** Representative micrographs of wild-type SOD-3::GFP and *ins-1(nj32)*; SOD-3::GFP animals shifted to OP50 + $H_2O$ ($H_2O$) or OP50 + 40 mM glucose (Glucose) plates 24 hours prior to imaging. **(G)** Quantification of intestinal SOD-3::GFP levels of wild-type and *ins-1 (nj32)* mutants shifted to OP50 + $H_2O$ ($H_2O$) or OP50 + 40 mM glucose (Glucose) plates 24 hours prior to imaging. The underlying numerical data can be found in S1 Data. CTCF, calculated total fluorescence; HS, heatshock; RFP, red fluorescent protein.
(EPS)

**S1 Table. Strains used in this study.** Strain names and genotypes of the *C. elegans* strains used in this study.
(DOCX)

**S1 Data. All the underlying numerical data and statistical analysis of this study.**
(XLSX)

**S1 Raw Image. Original western blot image for S3D Fig.**
(EPS)

**S1 Text. Supplementary methods.**
(DOCX)

## Acknowledgments

We thank Oliver Hobert, Mark Febbraio, Corey Laverty, and members of the Pocock Laboratory and for advice and comments on the manuscript. Some strains were provided by the *Caenorhabditis* Genetics Center (University of Minnesota), which is funded by NIH Office of Research Infrastructure Programs (P40 OD010440), and by the National BioResource Project of Japan. We extend our thanks to Chris Hopkins and Trisha Brock (Knudra Transgenics) for their expertise in genome-editing to generate CRISPR/Cas9 tagged ETS-5-GFP. We thank

Masahiro Tomioka for supplying the *rimb-1p*::*daf-2a cDNA* rescue line and plasmid and Ikue Mori for supplying the *ttx-3p*::*ins-1cDNA* construct.

## Author Contributions

**Conceptualization:** Ava Handley, Roger Pocock.

**Data curation:** Ava Handley, Roger Pocock.

**Formal analysis:** Ava Handley, Roger Pocock.

**Funding acquisition:** Roger Pocock.

**Investigation:** Ava Handley, Qiuli Wu, Tessa Sherry, Rebecca Cornell, Roger Pocock.

**Methodology:** Ava Handley, Qiuli Wu, Tessa Sherry, Rebecca Cornell, Roger Pocock.

**Project administration:** Roger Pocock.

**Resources:** Roger Pocock.

**Supervision:** Ava Handley, Roger Pocock.

**Validation:** Roger Pocock.

**Writing – original draft:** Ava Handley, Roger Pocock.

**Writing – review & editing:** Ava Handley, Qiuli Wu, Tessa Sherry, Rebecca Cornell, Roger Pocock.

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
