## [Editor Report · Decision Letter 0]

24 Nov 2021

Dear Dr Pocock, 

Thank you for submitting your manuscript entitled "Diet-responsive Transcriptional Regulation of Insulin in a Single Neuron Controls Systemic Metabolism" for consideration as a Research Article by PLOS Biology. Please accept my apologies for our delay in sending you an initial decision.

Your manuscript has now been evaluated by the PLOS Biology editorial staff as well as by an academic editor with relevant expertise and I am writing to let you know that we would like to send your submission out for external peer review.

Once your full submission is complete, your paper will undergo a series of checks in preparation for peer review. Once your manuscript has passed the checks it will be sent out for review. 

If your manuscript has been previously reviewed at another journal, PLOS Biology is willing to work with those reviews in order to avoid re-starting the process. Submission of the previous reviews is entirely optional and our ability to use them effectively will depend on the willingness of the previous journal to confirm the content of the reports and share the reviewer identities. Please note that we reserve the right to invite additional reviewers if we consider that additional/independent reviewers are needed, although we aim to avoid this as far as possible. In our experience, working with previous reviews does save time. 

If you would like to send your previous reviewer reports to us, please specify this in the cover letter, mentioning the name of the previous journal and the manuscript ID the study was given, and include a point-by-point response to reviewers that details how you have or plan to address the reviewers' concerns. Please contact me at the email that can be found below my signature if you have questions. 

Please re-submit your manuscript within two working days, i.e. by Nov 26 2021 11:59PM.

Kind regards,

Lucas

Lucas Smith

Associate Editor

PLOS Biology

lsmith@plos.org

---

## [Decision Letter · Decision Letter 1]

4 Feb 2022

Dear Dr Pocock,

I am writing on behalf of my colleague Dr Lucas Smith, who is currently on paternity leave.

Thank you for submitting your manuscript "Diet-responsive Transcriptional Regulation of Insulin in a Single Neuron Controls Systemic Metabolism" for consideration as a Research Article at PLOS Biology. Your manuscript has been evaluated by the PLOS Biology editors, by an Academic Editor with relevant expertise, and by three independent reviewers.

In light of the reviews (below), we will not be able to accept the current version of the manuscript, but we would welcome re-submission of a much-revised version that takes into account the reviewers' comments. We cannot make any decision about publication until we have seen the revised manuscript and your response to the reviewers' comments. Your revised manuscript is also likely to be sent for further evaluation by the reviewers. Please accept my apologies for the long delay in communicating this decision to you.

We expect to receive your revised manuscript within 3 months. 

**IMPORTANT - SUBMITTING YOUR REVISION**

Your revisions should address the specific points made by each reviewer. The Academic Editor thinks you should pay special attention to reviewer 1's major points 1, 2, and 3. These would need to be addressed with additional data. 

Please submit the following files along with your revised manuscript:

*Re-submission Checklist*

*Published Peer Review*

*PLOS Data Policy*

*Blot and Gel Data Policy*

Sincerely,

Gabriel Gasque on behalf of

Lucas Smith

Associate Editor

PLOS Biology

lsmith@plos.org

REVIEWS:

Reviewer #1, Seung-Jae V. Lee: Handley et al., 2021, PLOS Biology

In this manuscript entitled "Diet-responsive transcriptional regulation of insulin in a single neuron controls systemic metabolism", the authors showed that dietary glucose feeding from L4 stage significantly decreased ETS-5-mediated ins-1 expression in the BAG neurons and subsequently reduced intestinal daf-2 activity to increase DAF-16 expression. These series of reactions modulated foraging behavior and intestinal fat levels. This paper provides a new mechanistic insight into a discriminative role of insulin-like peptide in a single neuron and insulin/IGF-1 signaling in distal tissues under high dietary glucose conditions during post-developmental stage in C. elegans.

Major comment

1. The authors tested only one glucose concentration (40 mM) for glucose consumption. I wonder whether the downregulation of ins-1p::NLS::GFP by glucose is dose-dependent or not. 

2. In figures 4,5, and 6, for key experiments, the authors need to show the data with glucose. It will substantially increase the cohesiveness of the paper.

3. For key fat staining experiments, they need to obtain the data with L-glucose as well as D-glucose. If the fat staining data are similar between these two, glucose sensing or glycation mediate this signaling. If the data are different between these two, nutrients and metabolism are important, suggesting the existence of additional metabolic signaling for fat regulation in addition to the signaling axis shown in this paper. I think both data will be interesting and will help this paper. 

Minor comment

1. In Fig. 4C, the quantification numbers are not sufficient to show the effect of ins-1 mutations on concentration of TAGs. They can perform additional sets or move the data to supplementary figures.

2. In Fig. 6B, please add the data of the ins-1(nj32) single mutants at 25 °C. 

3. "As loss of ins-1 in the daf-2(e1370) mutant leads to an increase in fat storage at 20 °C, these data suggest that BAG-expressed INS-1 activates DAF-2, and when INS-1 is absent DAF-2 activity is decreased." is not consistent with the data in figure 4. There is no speculation about why ins-1(nj32) mutants displayed stronger phenotype for lipid accumulation or exploration than ins-1(tm1888) mutants. Please discuss this issue by describing the allelic difference between ins-1(nj32) and ins-1(tm1888) mutants.

5. Minor typos, grammatical and spacing errors throughout the manuscript need to be corrected. Following are some examples. 

Please add space between number and unit.

Please be consistent for "life span" and "lifespan".

Please be consistent for "antagonize" and "antagonise".

6. It seems sufficient to just write "mid-stage L4 animals" without using "Christmas-tree" for stage description.

7. In Fig. 2E, it will be better to arrange the bar graphs to be consistent with the order of results.

8. Please follow the C. elegans genetic nomenclature. For example, ins-1(nj32); daf-2(e1370) in Fig. 6 should be corrected to daf-2(e1370); ins-1(nj32).

Reviewer #2: In this manuscript, Handley et al uses C. elegans as a model system to address the role of insulin-like signaling in coordinating multiple physiological events such as fat storage and food-seeking behavior in response to excess dietary glucose. First, they found that excess dietary glucose decreases expression of ins-1 specifically in BAG sensory neurons and this regulation does not depend on neurotransmission or neuropeptide signaling and is maintained in che-3 mutants that are defective in cilia. Next, they showed that a transcription factor ETS-5 regulates ins-1 expression in response to excess glucose and the expression of ets-5 also decreases when dietary glucose level is high. They further showed that ets-5 binds to the promoter of ins-1 and thus causally linked these two molecular responses. Meanwhile they showed that ins-1 expression in BAG suppresses intestine fat storage and facilitates food-seeking movement. These results place ins-1 signaling in between external condition of high glucose and internal response of fat storage and food-seeking behavior. Next, they further showed that ins-1 signal from BAG antagonizes daf-2 receptor in the intestine to regulate fat storage and acts through daf-16 in the intestine in response to excess glucose. The results characterize the function of an insulin-like pathway in regulating fat metabolism and food seeking that contribute to our understanding of metabolic responses to glucose stress, a question that has important impacts on human health issues. 

Overall, the experiments are well designed and carefully conducted. The results are compelling. I have a few suggestions that may help to improve the manuscript. 

1. The authors pointed out that they used a lower glucose concentration and a shorter consumption duration than those in several other studies and that these differences may be important for the characterization of the signaling mechanisms in this study. It seems that the glucose concentration is still high, and it will be informative if authors can further discuss various impacts that different levels of excess glucose concentrations may have on physiological processes and perhaps how cells differentially respond to these external stresses.

2. Their experiment on POD-2 suggests that ins-1 signals to fat storage to regulate food seeking behavior. This is interesting and would be helpful if authors can discuss potential signaling pathways from intestinal fat storage to food-seeking movements. 

3. It is possible that excess glucose affects the growth of OP50 which subsequently affects worm metabolism. Their experiments using L-glucose argues against this possibility. But, it will be very helpful if authors discuss this possibility and how much they think that the external glucose exerts the effect on worm metabolism directly or indirectly. 

4. Mutants of cilia are often partially defective in sensing. It is not clear how much che-3 is defective in sensing and whether it is possible that ins-1 expression in BAG still partially depends on sensing external cues? 

5. On page 16, they stated that "Here, we found that DAF-16 is required for generating ~27 % of the fat accumulated over 24 hours on 40 mM glucose (Fig. 7C). This means that DAF-16 is not required for accumulating most (>70%) of the fat increase observed when animals are fed 40 mM glucose for 24 hours." How did authors quantify the effect here? It may be helpful to explain in the methods if they have not. 

6. How does L-glucose have a similar effect as D-glucose? They very briefly discussed about this. This effect is not expected but interesting. It will be helpful if the authors can further speculate the mechanism. 

7. In figS1D, it is very difficult to see what the authors are showing. 

Reviewer #3: Metabolic homeostasis is dependent on specific and conserved signaling pathways such as the insulin/insulin-like signaling pathway in metazoans. This work involves the use of C. elegans as a model to assess how exogenous glucose impacts signaling pathway in the sensory neurons. Authors convincingly determine:

* Dietary glucose reduces INS-1 expression in the BAG glutamatergic sensory neurons.

* INS-1 is controlled by the ETS-5 transcription factor which is also down regulated in response to dietary glucose.

* INS-1 acts from the BAG neurons and not other INS-1 expressing neurons to inhibit fat storage via DAF-2. 

Comments:

* Authors show a schematic diagram of what ins-1 expression is expected to be in figure 5A- it is unclear if this pattern was experimentally determined.

* INS-1 expressed in BAG neurons is thought to activate DAF-2 and when INS-1 is absent DAF-2 activity is decreased. This is shown genetically but unclear if this is what is occurring in a glucose fed animal. If authors could make this more clear it would strengthen the manuscript.

* Most experiments assess the INS-1, ETS-5, Insulin signaling pathway. Authors don't quite integrate how a glucose diet directly impacts fat storage/exploration behavior and if such is indeed occurring through the ins-1; ETS-5 pathways within BAG neurons. 

* Transition between concepts were sometimes disjointed (eg from the idea that ins-1 mutant spent more time in a quiescent state and relation to ets-5 fat levels, pg 14). Introduce questions or hypothesis to provide rationale for next experiment.

* Minor comment- document would be easier to review if the page number and line number are placed on the document. If using a space between a number and the % sign make sure it is non-breakable; I think "30%" is more common than "30 %" but defer to the editor.

* Fig 7D needs a scale bar

* Can authors address if it is indeed the glucose and not a byproduct produced by the bacterial metabolism of glucose (perhaps use delta-PTS strain to double check the results shown in Fig 1B with OP50; as these results are foundational to the study.

---

## [Decision Letter · Decision Letter 2]

25 Apr 2022

Dear Dr Pocock,

Thank you for submitting your revised Research Article entitled "Diet-responsive Transcriptional Regulation of Insulin in a Single Neuron Controls Systemic Metabolism" for publication in PLOS Biology. I have now obtained advice from the original reviewers and have discussed their comments with the Academic Editor. 

Based on the reviews, we will probably accept this manuscript for publication, provided you satisfactorily address the the data and other policy-related requests stated below.

We expect to receive your revised manuscript within two weeks. 

*Published Peer Review History*

*Press*

Sincerely,

Ines

--

Ines Alvarez-Garcia, PhD

Senior Editor

PLOS Biology

Fig. 1C; Fig. 2A-F; Fig. 3C, D, F, H; Fig. 4A, C-E; Fig. 5B-G; Fig. 6A, B, E; Fig. 7A-C, E, F; Fig. S1A, C, D; Fig. S3E; Fig. S4B; Fig. S5A, C, E and Fig. S6B, C, E, G

We require the original, uncropped and minimally adjusted images supporting all blot and gel results reported in an article's figures or Supporting Information files. We will require these files before a manuscript can be accepted so please prepare and upload them now. Please carefully read our guidelines for how to prepare and upload this data: https://journals.plos.org/plosbiology/s/figures#loc-blot-and-gel-reporting-requirements 

Reviewers' comments

Rev. 1: Seung-Jae V. Lee – note this reviewer has signed his review

The authors addressed my comments satisfactorily with further experiments and by modifying text.

Rev. 2:

In the revised manuscript, the authors fully addressed my questions and comments. I recommend this paper for publication.

Rev. 3:

Authors addressed all comments.

---

## [Editor Report · Decision Letter 3]

29 Apr 2022

Dear Dr Pocock,

On behalf of my colleagues and the Academic Editor, Heidi Tissenbaum, I am happy to say that we can in principle accept your Research Article entitled "Diet-responsive Transcriptional Regulation of Insulin in a Single Neuron Controls Systemic Metabolism" for publication in PLOS Biology, provided you address any remaining formatting and reporting issues. These will be detailed in an email that will follow this letter and that you will usually receive within 2-3 business days, during which time no action is required from you. Please note that we will not be able to formally accept your manuscript and schedule it for publication until you have completed any requested changes.

PRESS

Many congratulations and thank you again for choosing PLOS Biology for publication and supporting Open Access publishing. We look forward to publishing your study. 

Sincerely, 

Ines

--

Ines Alvarez-Garcia, PhD 

Senior Editor 

PLOS Biology
